# CHARACTERIZING AUDIO ADVERSARIAL EXAMPLES USING TEMPORAL DEPENDENCY

**Zhuolin Yang**
Shanghai Jiao Tong University

**Bo Li**
University of Illinois at Urbana–Champaign

**Pin-Yu Chen**
IBM Research

**Dawn Song**
UC, Berkeley

## ABSTRACT

Recent studies have highlighted adversarial examples as a ubiquitous threat to different neural network models and many downstream applications. Nonetheless, as unique data properties have inspired distinct and powerful learning principles, this paper aims to explore their potentials towards mitigating adversarial inputs. In particular, our results reveal the importance of using the temporal dependency in audio data to gain discriminate power against adversarial examples. Tested on the automatic speech recognition (ASR) tasks and three recent audio adversarial attacks, we find that (i) input transformation developed from image adversarial defense provides limited robustness improvement and is subtle to advanced attacks; (ii) temporal dependency can be exploited to gain discriminative power against audio adversarial examples and is resistant to adaptive attacks considered in our experiments. Our results not only show promising means of improving the robustness of ASR systems but also offer novel insights in exploiting domain-specific data properties to mitigate negative effects of adversarial examples.

## 1 INTRODUCTION

Deep Neural Networks (DNNs) have been widely adopted in a variety of machine learning applications (Krizhevsky et al., 2012; Hinton et al., 2012; Levine et al., 2016). However, recent work has demonstrated that DNNs are vulnerable to adversarial perturbations (Szegedy et al., 2014; Goodfellow et al., 2015). An adversary can add negligible perturbations to inputs and generate adversarial examples to mislead DNNs, first found in image-based machine learning tasks (Goodfellow et al., 2015; Carlini & Wagner, 2017a; Liu et al., 2017; Chen et al., 2017b;a; Su et al., 2018).

Beyond images, given the wide application of DNN-based audio recognition systems, such as Google Home and Amazon Alexa, audio adversarial examples have also been studied recently (Carlini & Wagner, 2018; Alzantot et al., 2018; Cisse et al., 2017; Kreuk et al., 2018). Comparing between image and audio learning tasks, although their state-of-the-art DNN architectures are quite different (i.e., convolutional v.s. recurrent neural networks), the attacking methodology towards generating adversarial examples is fundamentally unanimous - finding adversarial perturbations through the lens of maximizing the training loss or optimizing some designed attack objectives. For example, the same attack loss function proposed in (Cisse et al., 2017) is used to generate adversarial examples in both visual and speech recognition models. Nonetheless, different types of data usually possess unique or domain-specific properties that can potentially be used to gain discriminative power against adversarial inputs. In particular, the temporal dependency in audio data is an innate characteristic that has already been widely adopted in the machine learning models. However, in addition to improving learning performance on natural audio examples, it is still an open question on whether or not the temporal dependency can be exploited to help mitigate negative effects of adversarial examples.

The focus of this paper has two folds. First, we investigate the robustness of automatic speech recognition (ASR) models under *input transformation*, a commonly used technique in the image domain to mitigate adversarial inputs. Our experimental results show that four implemented transformation techniques on audio inputs, including waveform quantization, temporal smoothing, down-

sampling and autoencoder reformation, provide limited robustness improvement against the recent attack method proposed in (Athalye et al., 2018), which aims to circumvent the gradient obfuscation issue incurred by input transformations. Second, we demonstrate that temporal dependency can be used to gain discriminative power against adversarial examples in ASR. We perform the proposed temporal dependency method on both the LIBRIS (Graetz et al., 1986) and Mozilla Common Voice datasets against three state-of-the-art attack methods (Carlini & Wagner, 2018; Alzantot et al., 2018; Yuan et al., 2018) considered in our experiments and show that such an approach achieves promising identification of non-adaptive and adaptive attacks. Moreover, we also verify that the proposed method can resist strong proposed adaptive attacks in which the defense implementations are known to an attacker. Finally, we note that although this paper focuses on the case of audio adversarial examples, the methodology of leveraging unique data properties to improve model robustness could be naturally extended to different domains. The promising results also shed new lights in designing adversarial defenses against attacks on various types of data.

**Related work** An adversarial example for a neural network is an input $x_{adv}$ that is similar to a natural input $x$ but will yield different output after passing through the neural network. Currently, there are two different types of attacks for generating audio adversarial examples: the Speech-to-Label attack and the Speech-to-Text attack. The Speech-to-Label attack aims to find an adversarial example $x_{adv}$ close to the original audio $x$ but yields a different (wrong) label. To do so, Alzantot et al. proposed a genetic algorithm (Alzantot et al., 2018), and Cisse et al. proposed a probabilistic loss function (Cisse et al., 2017). The Speech-to-Text attack requires the transcribed output of the adversarial audio to be the same as the desired output, which has been made possible by Carlini and Wagner (Carlini & Wagner, 2018) using optimization-based techniques operated on the raw waveforms. Iter et al. leveraged extracted audio features called Mel Frequency Cepstral Coefficients (MFCCs) (Iter et al., 2017). Yuan et al. demonstrated the practical "wav-to-API" audio adversarial attacks (Yuan et al., 2018). Another line of research focuses on adversarial training or data augmentation to improve model robustness (Serdyuk et al., 2016; Michelsanti & Tan, 2017; Sriram et al., 2017; Sun et al., 2018), which is beyond our scope. Our proposed approach focuses on gaining the discriminative power against adversarial examples through embedded temporal dependency, which is compatible with any ASR model and does not require adversarial training or data augmentation.

## 2 Do Lessons from Image Adversarial Examples Transfer to Audio Domain?

Although in recent years both image and audio learning tasks have witnessed significant breakthroughs accomplished by advanced neural networks, these two types of data have unique properties that lead to distinct learning principles. In images, the pixels entail spatial correlations corresponding to hierarchical object associations and color descriptions, which are leveraged by the convolutional neural networks (CNN) for feature extraction. In audios, the waveforms possess apparent temporal dependency, which is widely adopted by the recurrent neural networks (RNNs). For the segmentation task in the image domain, spatial consistency has played an important role in improving model robustness (Lowe, 1999). However, it remains unknown whether temporal dependency can have a similar effect of improving model robustness against audio adversarial examples. In this paper, we aim to address the following fundamental questions: (a) *do lessons learned from image adversarial examples transfer to the audio domain?*; and (b) *can temporal dependency be used to discriminate audio adversarial examples?* Moreover, studying the discriminative power of temporal dependency in audios not only highlights the importance of using unique data properties towards building robust machine learning models but also aids in devising principles for investigating more complex data such as videos (spatial + temporal properties) or multimodal cases (e.g., images + texts).

Here we summarize two primary findings concluded from our experimental results in Section 4.

**Audio input transformation is not effective against adversarial attacks** Input transformation is a widely adopted defense technique in the image domain, owing to its low operating cost and easy integration with the existing network architecture (Luo et al., 2015; Wang et al., 2016; Dziugaite et al., 2016). Generally speaking, input transformation aims to perform certain feature transformation on the raw image in order to disrupt the adversarial perturbations before passing it to a neural network. Popular approaches include bit quantization, image filtering, image reprocessing, and autoencoder reformation (Xu et al., 2017; Guo et al., 2017; Meng & Chen, 2017). However, many existing

methods are shown to be bypassed by subsequent or adaptive adversarial attacks (Carlini & Wagner, 2017b; He et al., 2017; Carlini & Wagner, 2017c; Lu et al., 2018). Moreover, Athalye et al. (Athalye et al., 2018) has pointed out that input transformation may cause *obfuscated gradients* when generating adversarial examples and thus gives a false sense of robustness. They also demonstrated that in many cases this gradient obfuscation issue can be circumvented, making input transformation still vulnerable to adversarial examples. Similarly, in our experiments we find that audio input transformations based on waveform quantization, temporal filtering, signal downsampling or autoencoder reformation suffers from similar weakness: the tested model with input transformation becomes fragile to adversarial examples when one adopts the attack considering gradient obfuscation as in (Athalye et al., 2018).

**Temporal dependency possesses strong discriminative power against adversarial examples in automatic speech recognition** Instead of input transformation, in this paper, we propose to exploit the inherent temporal dependency in audio data to discriminate adversarial examples. Tested on the automatic speech recognition (ASR) tasks, we find that the proposed methodology can effectively detect audio adversarial examples while minimally affecting the recognition performance on normal examples. In addition, experimental results show that a considered adaptive adversarial attack, even when knowing every detail of the deployed temporal dependency method, cannot generate adversarial examples that bypass the proposed temporal dependency-based approach.

Combining these two primary findings, we conclude that the weakness of defense techniques identified in the image case is very likely to be transferred to the audio domain. On the other hand, exploiting unique data properties to develop defense methods, such as using temporal dependency in ASR, can lead to promising defense approaches that can resist adaptive adversarial attacks.

## 3 INPUT TRANSFORMATION AND TEMPORAL DEPENDENCY IN AUDIO DATA

In this section, we will introduce the effect of basic input transformations on audio adversarial examples, and analyze temporal dependency in audio data. We will also show that such temporal dependency can be potentially leveraged to discriminate audio adversarial examples.

### 3.1 AUDIO ADVERSARIAL EXAMPLES UNDER SIMPLE INPUT TRANSFORMATIONS

Inspired by image input transformation methods and as a first attempt, we applied some primitive signal processing transformations to audio inputs. These transformations are useful, easy to implement, fast to operate and have delivered several interesting findings.

**Quantization:** By rounding the amplitude of audio sampled data into the nearest integer multiple of $q$, the adversarial perturbation could be disrupted since its amplitude is usually small in the input space. We choose $q = 128, 256, 512, 1024$ as our parameters.

**Local smoothing:** We use a sliding window of a fixed length for local smoothing to reduce the adversarial perturbation. For an audio sample $x_i$, we consider the $K - 1$ samples before and after it, denoted by $[x_{i-K+1}, \ldots, x_i, \ldots, x_{i+K-1}]$, as a local reference sequence and replace $x_i$ by the smoothed value (average, median, etc) of its reference sequence.

**Downsampling:** Based on sampling theory, it is possible to down-sample a band-limited audio file without sacrificing the quality of the recovered signal while mitigating the adversarial perturbations in the reconstruction phase. In our experiments, we down-sample the original 16kHz audio data to 8kHz and then perform signal recovery.

**Autoencoder:** In adversarial image defending field, the MagNet defensive method (Meng & Chen, 2017) is an effective way to remove adversarial noises: Implement an autoencoder to project the adversarial input distribution space into the benign distribution. In our experiments, we implement a sequence-to-sequence autoencoder and the whole audio will be cut into frame-level pieces passing through the autoencoder and concatenate them in the final stage, while using the whole audio passing the autoencoder directly is proved to be ineffective and hard to utilize the underlying information.

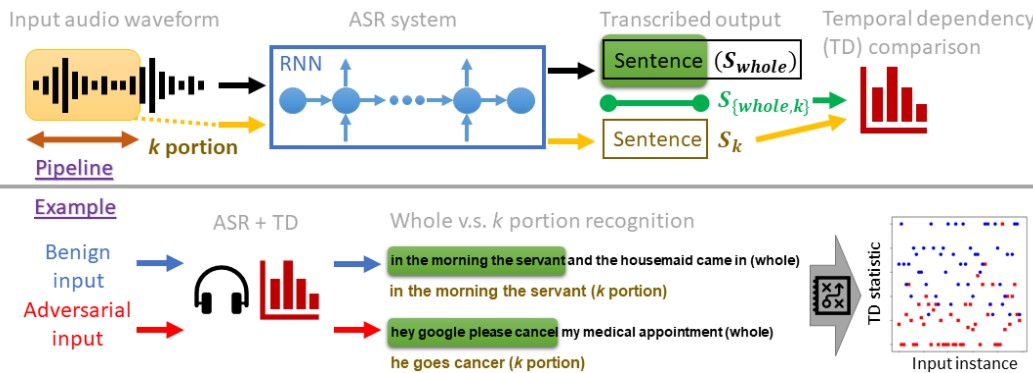

Figure 1: Pipeline and example of the proposed temporal dependency (TD) based method for discriminating audio adversarial examples.

## 3.2 TEMPORAL DEPENDENCY BASED METHOD (TD)

Due to the fact that audio sequence has an explicit temporal dependency (e.g., correlations in consecutive waveform segments), here we aim to explore if such temporal dependency will be affected by adversarial perturbations. The pipeline of the temporal dependency based method is shown in Figure 1. Given an audio sequence, we propose to select the first $k$ portion of it (i.e., the prefix of length $k$) as input for ASR to obtain transcribed results as $S_k$. We will also insert the whole sequence into ASR and select the prefix of length $k$ of the transcribed result as $S_{\{whole,k\}}$, which has the same length as $S_k$. We will then compare the consistency between $S_k$ and $S_{\{whole,k\}}$ in terms of temporal dependency distance. Here we adopt the word error rate (WER) as the distance metric (Levenshtein, 1966). For normal/benign audio instance, $S_k$ and $S_{\{whole,k\}}$ should be similar since the ASR model is consistent for different sections of a given sequence due to its temporal dependency. However, for audio adversarial examples, since the added perturbation aims to alter the ASR output toward the targeted transcription, it may fail to preserve the temporal information of the original sequence. Therefore, due to the loss of temporal dependency, $S_k$ and $S_{\{whole,k\}}$ , in this case, will not be able to produce consistent results. Based on such hypothesis, we leverage the prefix of length $k$ of the transcribed results and the transcribed $k$ portion to potentially recognize adversarial inputs.

## 4 EXPERIMENTAL RESULTS

The presentation flows of the experimental results are summarized as follows. We will first introduce the datasets, target learning models, attack methods, and evaluation metrics for different defense/detection methods that we focus on. We then discuss the defense/detection effectiveness for different methods against each attack respectively. Finally, we evaluate strong adaptive attacks against these defense/detection methods. We show that due to different data properties, the autoencoder based defense cannot effectively recover the ground truth for adversarial audios and may also have negative effects on benign instances as well. Input transformation is less effective in defending adversarial audio than images. In addition, even when some input transformation is effective for recovering some adversarial audio data, we find that it is easy to perform adaptive attacks against them. The proposed TD method can effectively detect adversarial audios generated by different attacks targeting on various learning tasks (classification and speech-to-text translation). In particular, we propose different types of strong adaptive attacks against the TD detection method. We show that these strong adaptive attacks are not able to generate effective adversarial audio against TD and we provide some case studies to further understand the performance of TD.

### 4.1 EXPERIMENTAL SETUP

In our experiments, we measure the effectiveness on several adversarial audio generation methods. For audio classification attack, we used Speech Commands dataset. For the speech-to-text attack, we benchmark each method on both LibriSpeech and Mozilla Common Voice dataset. In particular, for the Commander Song attack (Yuan et al., 2018), we measure the generated adversarial audios given by the authors.

**Dataset** *LibriSpeech dataset*: LibriSpeech (Panayotov et al., 2015) is a corpus of approximately 1000 hours of 16Khz English speech derived from audiobooks from the LibriVox project. We used samples from its test-clean dataset in their website and the average duration is 4.294s. We generated adversarial examples using the attack method in (Carlini & Wagner, 2018).

*Mozilla Common Voice dataset*: Common Voice is a large audio dataset provided by Mozilla. This dataset is public and contains samples from human speaking audio files. We used the 16Khz-sampled data released in (Carlini & Wagner, 2018), whose average duration is 3.998s. The first 100 samples from its test dataset are used to mount attacks, which is the same attack experimental setup as in (Carlini & Wagner, 2018).

Speech Commands dataset: Speech Commands dataset (Warden, 2018) is an audio dataset contains 65000 audio files. Each audio is just a single command lasting for one second. Commands are "yes", "no", "up", "down", "left", "right", "on", "off", "stop", and "go".

**Model and learning tasks** For the speech-to-text task, we use DeepSpeech speech-to-text transcription network, which is a biRNN based model with beam search to decode text. For audio classification task, we use a convolutional speech commands classification model. For the Command Song attack, we evaluate the performance on Kaldi speech recognition platform.

**Attack Methods**

*Genetic algorithm based attack against audio classification (GA)*: For the audio classification task, we consider the state-of-the-art attack proposed in (Alzantot et al., 2018). Here an audio classification model is attacked and the audio classes include "yes, no, up, down, etc.". They aimed to attack such a network to misclassify an adversarial instance based on either targeted or untargeted attack.

*Commander Song attack against speech-to-text translation (Commander)*: Commander Song (Yuan et al., 2018) is a speech-to-text targeted attack which can attack audio extracted from popular songs. The adversarial audio can even be played over the air with its adversarial characteristics. Since the Commander Song codes are not available, we measure the effectiveness of the generated adversarial audios given by the authors.

*Optimization based attack against speech-to-text translation (Opt)*: We consider the targeted speech-to-text attack proposed by (Carlini & Wagner, 2018), which uses CTC-loss in a speech recognition system as an objective function and solves the task of adversarial attack as an optimization problem.

**Evaluation Metrics** For *defense method* such as input transformation, since it aims to recover the ground truth (original instances) from adversarial instances, we use the word error rate (WER) and character error rate (CER) (Levenshtein, 1966) as evaluation metrics to measure the recovery efficiency. WER and CER are commonly used metrics to measure the error between recovered text and the ground truth in word level or character level. Generally speaking, the error rate (ER) is defined by $ER = \frac{S+D+I}{N}$, where $S, D, I$ is the number of substitutions, deletions and insertions calculated by dynamic string alignment, and $N$ is the total number of word/character in the ground truth text.

To fairly evaluate the effectiveness of these transformations against speech-to-text attack, we also report the ratio of translation distance between instance and corresponding ground truth before and after transformation. For instance, as a controlled experiment, given an audio instance $x$ (adversarial instance is denoted as $x_{adv}$), its corresponding ground truth $y$, and the ASR function $g(\cdot)$, we calculate the effectiveness ratio for benign instances as $R_{benign} = \frac{D(g(T(x)),y)}{D(g(x),y)}$, where $T(\cdot)$ denotes the result of transformation and $D(\cdot, \cdot)$ characterizes the distance function (WER and CER in our case). For adversarial audio, we calculate the similar efficiency ratio as $R_{adv} = \frac{D(g(T(x_{adv})),y)}{D(g(x_{adv}),y)}$.

For the *detection method*, the standard evaluation metric is the area under curve (AUC) score, aiming to evaluate the detection efficiency. The proposed TD method is the first data-specific metric to detect adversarial audio, which focuses on how many adversarial instances are captured (true positive) without affecting benign instances (false positive). Therefore, we follow the standard criteria and report AUC for TD. For the proposed TD method, we compare the temporal dependency based on WER, CER, as well as the longest common prefix (LCP). LCP is a commonly used metric to evaluate the similarity between two strings. Given strings $b_1$ and $b_2$, the corresponding LCP is defined as $max_{b_1[:k]=b_2[:k]} k$, where $[:k]$ represents the prefix of length $k$ of a translated sentence.

Table 1: List of adversarial audio based attacks and corresponding evaluation results for defense and detection methods

| Learning tasks | | Classification | Speech-to-Text | | |
|---|---|---|---|---|---|
| Attack methods | | Genetic Algorithm (GA) | CommanderSong (Commander) | Opt. attack (Opt) | |
| Datasets | | SpeechCommand | Some popular songs | LibriSpeech | CommonVoice |
| Evaluation metrics of defense | | Average attack success rate | Target command recognition rate | Efficiency ratio $R_{benign}/R_{adv}$ | |
| Defense Method | Without defense | 84% | 100% | 1.0 / 1.0 | 1.0 / 1.0 |
| | Input trans. Down Samp. | 3.2% | 8% | 3.87 / 0.40 | 1.13 / 0.73 |
| | Input trans. Quan-256 | 2.1% | 4% | 1.13 / 0.28 | 1.56 / 0.74 |
| | Input trans. Median-4 | 2.7% | 4% | 1.18 / 0.34 | 0.98 / 0.77 |
| | Autoencoder | 8.2% | - | 9.84 / 0.97 | 2.09 / 0.80 |
| Evaluation metrics for detection | | - | Detection rate | AUC score | |
| Detection results of TD Method | | - | **1.00** | **0.930** | **0.936** |

## 4.2 Evaluation of Defense methods against adversarial audio

In this section, we measured our defense method of autoencoder based defense and input transformation defense for classification attack (GA) and speech-to-text attack (Commander and Opt). We summarize our work in Table 1 and list some basic results. For Commander, due to unreleased training data, we are not able to train an autoencoder. For GA and Opt we have sufficient data to train autoencoder.

**Input Transformation as Defense**

Here we perform the primitive input transformation for audio classification targeted attacks and evaluate the corresponding effects. Due to the space limitation, we defer the results of untargeted attacks to the supplemental materials.

*GA* We first evaluate our input transformation against the audio classification attack (GA) in (Alzantot et al., 2018). We implemented their attack with 500 iterations and limit the magnitude of adversarial perturbation within 5 (smaller than the quantization we used in transformation) and generated 50 adversarial examples per attack task (more targets are shown in the supplementary material). The attack success rate is 84% on average. For the ease of illustration, we use Quantization-256 as our input transformation. As observed in Figures 2 and 3, the attack success rates decreased to only 2.1%, and 63.8% of the adversarial instances have been converted back to their original (true) label. We also measure the possible effects on original audio due to our transformation methods: the original audio classification accuracy without our transformation is 89.2%, and the rate slightly decreased to 89.0% after our transformation, which means the effects of input transformation on benign instances are negligible. In addition, it also shows that for classification tasks, such input transformation is more effective in mitigating the negative effects of adversarial perturbation. This potential reason could be that classification tasks do not rely on audio temporal dependency but focus on local features, while speech-to-text task will be harder to defend based on the tested input transformations.

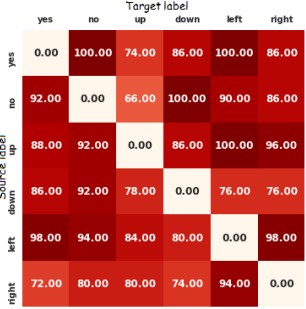

Figure 2: Attack success rates (%)

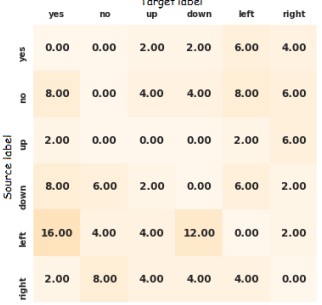

Figure 3: Attack success (%) after transformation

*Commander* We also evaluate our input transformation method against the Commander Song attack (Yuan et al., 2018), which implemented an Air-to-API adversarial attack. In the paper, the authors reported 91% attack detection rate using their defense method. We measured our Quan-256 input transformation on 25 adversarial examples obtained via personal communications. Based on

the same detection evaluation metric in (Yuan et al., 2018)[1], Quan-256 attains 100% detection rate for characterizing all the adversarial examples.

***Opt*** Here we consider the state-of-the-art audio attack proposed in (Carlini & Wagner, 2018). We separately choose 50 audio files from two audio datasets (Common Voice, LIBRIS) and generate attacks based on the CTC-loss. We evaluate several primitive signal processing methods as input transformation under WER and CER metrics in Table A1 and A2. We then also evaluate the WER and CER based effectiveness ratio we mentioned before to Quantify the effectiveness of transformation. $R_{benign}$ are shown in the brackets for the first two columns in Table A1 and A2, while $R_{adv}$ is shown in the brackets of last two columns within those tables. We compute our results using both ground truth and adversarial target "This is an adversarial example" as references.

Here small $R_{benign}$ which is close to 1 indicates that transformation has little effect on benign instances, small $R_{adv}$ represents transformation is effective recovering adversarial audio back to benign. From Tables A1 and A2 we showed that most of the input transformations (e.g., Median-4, Downsampling and Quan-256) effectively reduce the adversarial perturbation without affecting the original audio too much.

Although these input transformations show certain effectiveness in defending against adversarial audios, we find that it is still possible to generate adversarial audios by adaptive attacks in Section 4.4.

**Autoencoder as Defense**

Towards defending against (non-adaptive) adversarial images, MagNet (Meng & Chen, 2017) has achieved promising performance by using an autoencoder to mitigate adversarial perturbation. Inspired by it, here we apply a similar autoencoder structure for audio and test if such input transformation can be applied to defending against adversarial audio. We apply a MagNet-like method for feature-extracted audio spectrum map: we build an encoder to compress the information of origin audio features into latent vector $z$, then use $z$ for reconstruction by passing through another decoder network under frame level and combine them to obtain the transformed audio (Hsu et al., 2017). Here we analyzed the performance of Autoencoder transformation in both *GA* and *Opt* attack. We find that MagNet which gained great effectiveness on defending adversarial images in the oblivious attack setting (Carlini & Wagner, 2017c; Lu et al., 2018), has limited effect on the audio defense.

***GA*** We presented our results in Table 1 that against classification attack, Autoencoder did not perform well by only reducing attack success rate to $8.2\%$ defeat by other input transformation methods. Since you can reduce the attack success rate to $10\%$ by just destroying the origin audio data and altering to random guess, it's hard to say that Autoencoder method has good performance.

***Opt*** We report that the autoencoder works not very well for transforming benign instances (57.6 WER in Common Voice compared to 27.5 WER without transformation, 30.0 WER in LIBRIS compared to 12.4 WER without transformation), also fails to recover adversarial audio (76.5 WER in Common Voice and 99.4 WER in LIBRIS). This shows that the non-adaptive additive adversarial perturbation can bypass the MagNet-like autoencoder on audio, which implies different robustness implications of image and audio data.

### 4.3 Evaluation of TD detection method against Adversarial audio

In this section, we will evaluate the proposed TD detection method on different attacks. We will first report the AUC for detecting different attacks with TD to demonstrate the effectiveness, and we will provide some additional analysis and examples to help better understand TD. We only evaluate our TD method on speech-to-text attacks (Commander and Opt) because of the audio in the Speech Commands dataset for classification attack is just a single command lasting for one second and thus its temporal dependency is not obvious.

***Commander*** In Commander Song attack, we directly examine whether the generated adversarial audio is consistent with its prefix of length $k$ or not. We report that by using TD method with $k = \frac{1}{2}$, all the generated adversarial samples showed inconsistency and thus were successfully detected.

***Opt*** Here we show the empirical performance of distinguishing adversarial audios by leveraging the temporal dependency of audio data. In the experiments, we use these three metrics, WER,

---

[1]The authors set the detection threshold to be 0 and we used the same setting here.

CER and LCP, to measure the inconsistency between $S_k$ and $S_{\{whole,k\}}$. As a baseline, we also directly train a one layer LSTM with 64 hidden feature dimensions based on the collected adversarial and benign audio instances for classification. Some examples of translated results for benign and adversarial audios are shown in Table 2. Here we consider three types of adversarial targets: short – *hey google*; medium – *this is an adversarial example*; and long – *hey google please cancel my medical appointment*. We report the AUC score for these detection results for $k = 1/2$ in Table 3.

Table 2: Examples of the temporal dependency based detection method

| Type | Transcribed results |
|------|---------------------|
| Original | then good bye said the rats and they went home |
| the first half of Original | then good bye said the raps |
| | |
| Adversarial (short) | hey google |
| First half of Adversarial | he is |
| Adversarial (medium) | this is an adversarial example |
| First half of Adversarial | thes on adequate |
| Adversarial (long) | hey google please cancel my medical appointment |
| First half of Adversarial | he goes cancer |

We can see that by using WER as the detection metric, the temporal dependency based method can achieve AUC as high as 0.936 on Common Voice and 0.93 on LIBRIS. We also explore different values of $k$ and we observe that the results do not vary too much (detailed results can be found in Table A6 in Appendix). When $k = 4/5$, the AUC score based on CER can reach $0.969$, which shows that such temporal dependency based method is indeed promising in terms of distinguishing adversarial instances. Interestingly, these results suggest that the temporal dependency based method would suggest an easy-implemented but effective method for characterizing adversarial audio attacks.

### 4.4 ADAPTIVE ATTACKS AGAINST DEFENSE AND DETECTION METHODS

In this section, we measured some adaptive attack against the defense and detection methods. Since the autoencoder based defense almost fails to defend against different attacks, here we will focus on the input transformation based defense and TD detection. Given that Opt is the strongest attack here, we will mainly apply Opt to perform adaptive attack against the speech-to-text translation task. We list our experiments' structure in Table 4. For full results please refer to the Appendix.

**Adaptive Attacks Against Input Transformations** Here we apply adaptive attacks against the preceding input transformations and therefore evaluate the robustness of the input transformation as defenses. We implemented our adaptive attack based on three input transformation methods: Quantization, Local smoothing, and Downsampling. For these transformations, we leverage a gradient-masking aware approach to generate adaptive attacks.

In the optimization based attack (Carlini & Wagner, 2018), the attack achieved by solving the optimization problem: $\min_\delta \|\delta\|_2^2 + c \cdot l(x + \delta, t)$, where $\delta$ is referred to the perturbation, $x$ the benign audio, $t$ the target phrase, and $l(\cdot)$ the CTC-loss. Parameter $c$ is iterated to trade off the importance of being adversarial and remaining close to the original instance.

For quantization transformation, we assume the adversary knows the quantization parameter $q$. We then change our attack targeted optimization function to $\min_\delta \|q\delta\|_2^2 + c \cdot l(x + q\delta, t)$. After that, all the adversarial audios can be resistant against quantization transformations and it only increased a small magnitude of adversarial perturbation, which can be ignored by human ears. When $q$ is large enough, the distortion would increase but the transformation process is also ineffective due to too much information loss.

For downsampling transformation, the adaptive attack is conducted by performing the attack on the sampled elements of origin audio sequence. Since the whole process is differentiable, we can do adaptive attack through gradient directly and all the adversarial audios are able to attack.

For local smoothing transformation, it is also differentiable in case of average smoothing transformation, so we can pass the gradient effectively. To attack against median smoothing transformation, we can just convert the gradient back to the median and update its value, which is similar to the max-pooling layer's backpropagation process. By implementing the adaptive attack, all the smoothing transformation is shown to be ineffective.

Table 3: AUC results of the proposed temporal dependency method

| Dataset | LSTM | TD (WER) | TD (CER) | TD (LCP ratio) |
|---|---|---|---|---|
| Common Voice | 0.712 | **0.936** | 0.916 | 0.859 |
| LIBRIS | 0.645 | 0.930 | **0.933** | 0.806 |

Table 4: Evaluation of adaptive attacks

| Attack methods | | | Optimization based attack (Opt) | |
|---|---|---|---|---|
| Datasets | | | LibriSpeech | CommonVoice |
| Evaluation metrics of adaptive attack | | | Attack success rate | |
| Defense Method | Input trans. | Down Samp. | 92% | 90% |
| | | Quan-256 | 98% | 100% |
| | | Median-4 | 98% | 96% |
| Evaluation metrics for detection | | | AUC score | |
| Detection results of TD Method | | | 0.930 | 0.936 |
| Segment attack | | | 2% success rate | 2% success rate |
| Concatenation attack | | | Failed. | Failed. |
| Combination attack under both random $k_A$ and $k_D$ | | | 0.873 | 0.877 |

We chose our samples randomly from LIBRIS and Common Voice audio dataset with 50 audio samples each. We implemented our adaptive attack on the samples and passed them through the corresponding input transformation. We use down-sampling from 16kHZ to 8kHZ, median / average smoothing with one-sided sequence length $K = 4$, quantization method with $q = 256$ as our input transformation methods. In (Carlini & Wagner, 2018), Decibels (a logarithmic scale that measures the relative loudness of an audio sample) is applied as the measurement of the magnitude of perturbation: $dB(x) = \max_i 20 \cdot \log_{10}(x_i)$, which $x$ referred as adversarial audio sampled sequence. The relative perturbation is calculated as $dB_x(\delta) = dB(\delta) - dB(x)$, where $\delta$ is the crafted adversarial noise.

We measured our adaptive attack based on the same criterion. We show that all the adaptive attacks become effective with reasonable perturbation, as shown in Table 6. As suggested in (Carlini & Wagner, 2018), almost all the adversarial audios have distortion $dB_x(\delta)$ from -15dB to -45dB which is tolerable to human ears. From Table 6, the added perturbation are mostly within this range.

**Adaptive Attacks Against Temporal Dependency Based Method** To thoroughly evaluate the robustness of temporal dependency based method, we also perform some strong adaptive attack against it. Notably, even if the adversary knows $k$, the adaptive attack is hard to conduct due to the fact that this process is non-differentiable. Therefore, we propose three types of strong adaptive attacks here aiming to explore the robustness of the temporal consistency based method.

*Segment attack*: Given the knowledge of $k$, we first split the audio into two parts: the prefix of length $k$ of the audio $S_k$ and the rest $S_{k-}$. We then apply a similar attack to add perturbation to only $S_k$. We hope this audio can be attacked successfully without changing $S_{k-}$ since the second part would not receive gradient updates. Therefore, when performing the temporal-based consistency check, $T(S_k)$ would be translated consistently with $T(S_{\{whole,k\}})$.

*Concatenation attack*: To maximally leverage the information of $k$, here we propose two ways to attack both $S_k$ and $S_{k-}$ individually, and then concatenate them together.

**1.** the target of $S_k$ is the first $k-$portion of the adversarial target, and $S_{k-}$ is attacked to the rest.

**2.** the target of $S_k$ is the whole adversarial target, while we attack $S_{k-}$ to be silence, which means $S_{k-}$ transcribing nothing. This is different from the segment attack where $S_{k-}$ is not modified at all.

*Combination attack*: To balance the attack success rate for both sections and the whole sentence against TD, we apply the attack objective function as $\min_\delta \|\delta\|_2^2 + c \cdot (l(x + \delta, t) + l((x + \delta)_k, t_k))$, where $x$ refers to the whole sentence.

For segment attack, we found that in most cases the attack cannot succeed, that the attack success rate remains at 2% for 50 samples in both LIBRIS and Common Voice datasets, and some of the

Table 5: AUC of detecting Combination Attack based on TD method

| Combination Attack | Detection Parameter $k_D$ | TD metrics | | |
|---|---|---|---|---|
| | | WER | CER | LCP |
| $k_A = \{\frac{1}{2}\}$ | 1/2 | 0.607 | 0.518 | 0.643 |
| | 2/3 | 0.957 | 0.965 | 0.881 |
| | Rand(0.2, 0.8) | 0.889 | 0.882 | 0.776 |
| $k_A = \{\frac{1}{2}, \frac{2}{3}, \frac{3}{4}\}$ | 1/2 | 0.665 | 0.682 | 0.604 |
| | 2/3 | 0.653 | 0.664 | 0.564 |
| | 3/4 | 0.633 | 0.653 | 0.601 |
| | Rand(0.2, 0.8) | 0.785 | 0.832 | 0.642 |

Table 6: The $dB_x(\delta)$ evaluation of adaptive attack

| Dataset | Non-adaptive | Downsample | Quantization-256 | Median-4 | Average-4 |
|---|---|---|---|---|---|
| LIBRIS | -36.06 | -21.42 | -11.02 | -23.58 | -25.64 |
| CommmonVoice | -35.65 | -20.91 | -9.48 | -23.42 | -25.12 |

examples are shown in Appendix. We conjecture the reasons as: 1. $S_k$ alone is not enough to be attacked to the adversarial target due to the temporal dependency; 2. the speech recognition results on $S_{k-}$ cannot be applied to the whole recognition process and therefore break the recognition process for $S_k$.

For concatenation attack, we also found that the attack itself fails. That is, the transcribed result of $adv(S_k)+adv(S_{k-})$ differs from the translation result of $S_k+S_{k-}$. Some examples are shown in Appendix. The failure of the concatenation adaptive attack more explicitly shows that the temporal dependency plays an important role in audio. Even if the separate parts are successfully attacked into the target, the concatenated instance will again totally break the perturbation and therefore render the adaptive attack inefficient. On the contrary, such concatenation will have negligible effects on benign audio instances, which provides a promising direction to detect adversarial audio.

For combination attack, we vary the section portion $k_D$ used by TD and evaluate the cases where the adaptive attacker uses the same/different section $k_A$. We define Rand(a,b) as uniformly sampling from [a,b]. We consider a stronger attacker, for whom the $k_A$ can be a set containing random sections. The detection results for different settings are shown in Table 5. From the results, we can see that when $|k_A| = 1$, if the attacker uses the same $k_A$ as $k_D$ to perform the adaptive attack, the attack can achieve relatively good performance and if the attacker uses different $k_A$, the attack will fail with AUC above 85%. We also evaluate the case that defender randomly sample $k_D$ during the detection and find that it's very hard for the adaptive attacker to perform attacks, which can improve model robustness in practice. For $|k_A| > 1$, the attacker can achieve some attack success when the set contains $k_D$. But when $|k_A|$ increases, the attacker's performance becomes worse. The complete results are given in the Appendix. Notably, the random sample based TD appears to be robust in all cases.

## 5 CONCLUSION

This paper proposes to exploit the temporal dependency property in audio data to characterize audio adversarial examples. Our experimental results show that while four primitive input transformations on audio fail to withstand adaptive adversarial attacks, temporal dependency is shown to be resistant to these attacks. We also demonstrate the power of temporal dependency for characterizing adversarial examples generated by three state-of-the-art audio adversarial attacks. The proposed method is easy to operate and does not require model retraining. We believe our results shed new lights in exploiting unique data properties toward adversarial robustness.

## ACKNOWLEDGEMENT

This work is partially supported by DARPA grant 00009970.

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

# APPENDIX

## 5.1 RESULTS ON "AUTOENCODER TRANSFORMATION METHOD FOR SPEECH-TO-TEXT ATTACK" AND "PRIMITIVE TRANSFORMATION FOR SPEECH-TO-TEXT ATTACK"

Table A1: Evaluation on Common Voice with language model

| Transformation Methods | OriginWER(%) | OriginCER(%) | AdvWER(%) | AdvCER(%) |
|---|---|---|---|---|
| Without transformations | 27.5 | 14.3 | 95.9 | 80.1 |
| Autoencoder | 57.6 (2.09) | 34.1 (2.38) | 76.5 (0.80) | 49.8 (0.62) |
| Median-4 | 27.0 (0.98) | 14.6 (1.02) | 73.6 (0.77) | 42.4 (0.53) |
| Downsample | 31.2 (1.13) | 17.6 (1.23) | 69.6 (0.73) | 41.2 (0.51) |
| Quan-128 | 34.4 (1.25) | 21.3 (1.49) | 75.9 (0.79) | 45.3 (0.57) |
| Quan-256 | 42.9 (1.56) | 26.7 (1.87) | 70.7 (0.74) | 41.8 (0.52) |
| Quan-512 | 52.4 (1.90) | 37.1 (2.59) | 68.5 (0.71) | 45.0 (0.56) |
| Quan-1024 | 62.4 (2.27) | 47.2 (3.3) | 70 (0.73) | 51.2 (0.64) |

Table A2: Evaluation on LIBRIS with language model

| Transformation Methods | OriginWER(%) | OriginCER(%) | AdvWER(%) | AdvCER(%) |
|---|---|---|---|---|
| Without transformations | 3.05 | 1.46 | 102.8 | 86.5 |
| Autoencoder | 30.0 (9.84) | 15.1 (10.34) | 99.4 (0.97) | 58.1 (0.67) |
| Median-4 | 3.6 (1.18) | 1.7 (1.16) | 35.1 (0.34) | 19.0 (0.22) |
| Downsample | 11.8 (3.87) | 5.7 (3.90) | 41.2 (0.40) | 21.8 (0.25) |
| Quan-128 | 3.2 (1.04) | 1.5 (1.03) | 49.7 (0.48) | 28.2 (0.33) |
| Quan-256 | **3.5 (1.13)** | **1.7 (1.16)** | **29.1 (0.28)** | **15.4 (0.18)** |
| Quan-512 | 12.0 (3.93) | 6.6 (4.52) | 25.1 (0.24) | 13.3 (0.15) |
| Quan-1024 | 30.7 (10.06) | 20.3 (13.90) | 36.6 (0.36) | 24.1 (0.28) |

Table A3: Evaluation on Common Voice without passing through language model

| Transformation Methods | OriginWER(%) | OriginCER(%) | AdvWER(%) | AdvCER(%) |
|---|---|---|---|---|
| Without transformations | 37.7 | 18.5 | 95.8 | 83.0 |
| Median-4 | 43.4 (1.15) | 20.4 (1.10) | 83.0 (0.87) | 46.5 (0.56) |
| Down sampling | 47.2 (1.25) | 23.3 (1.26) | 77.6 (0.81) | 43.9 (0.53) |
| Quantization-128 | 47.3 (1.25) | 25.7 (1.39) | 80.7 (0.84) | 49.0 (0.59) |
| Quantization-256 | 52.5 (1.39) | 29.2 (1.58) | 73.4 (0.77) | 43.6 (0.53) |
| Quantization-512 | 64.1 (1.70) | 37.5 (2.03) | 73.7 (0.77) | 44.2 (0.53) |
| Quantization-1024 | 72.1 (1.91) | 50.4 (2.72) | 76.9 (0.80) | 53.0 (0.64) |

Table A4: Evaluation on LIBRIS without passing through language model

| Transformation Methods | OriginWER(%) | OriginCER(%) | AdvWER(%) | AdvCER(%) |
|---|---|---|---|---|
| Without transformations | 12.4 | 7.05 | 105.3 | 91.7 |
| Median-4 | 16.4 (1.32) | 8.0 (1.13) | 57.9 (0.55) | 27.5 (0.30) |
| Downsample | 24.2 (1.95) | 13.0 (1.84) | 60.9 (0.58) | 31.2 (0.34) |
| Quantization-128 | 13.4 (1.08) | 7.6 (1.08) | 66.1 (0.63) | 37.1 (0.40) |
| Quantization-256 | **16.3 (1.31)** | **8.9 (1.26)** | **48.6 (0.46)** | **24.0 (0.26)** |
| Quantization-512 | 27.5 (2.21) | 13.8 (1.96) | 47.0 (0.45) | 23.0 (0.25) |
| Quantization-1024 | 46.8 (3.77) | 25.4 (3.60) | 52.3 (0.50) | 30.0 (0.33) |

## 5.2 MORE RESULTS ON PRIMITIVE TRANSFORMATION METHOD FOR AUDIO CLASSIFICATION ATTACK

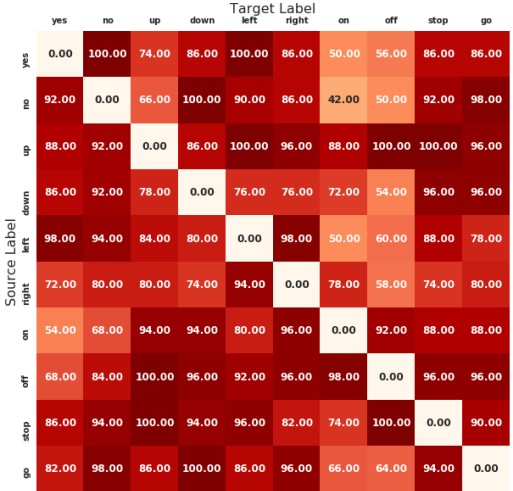

Figure A1: Successful attack rates

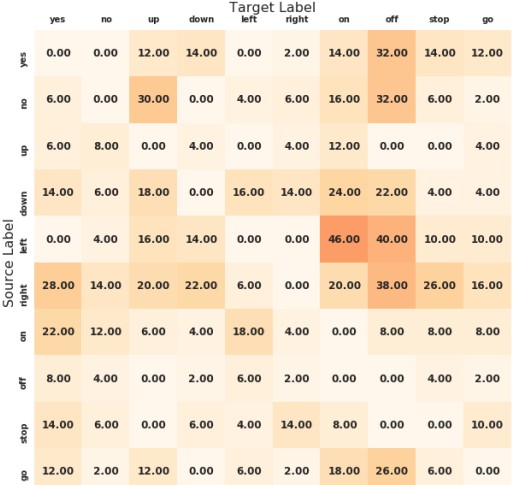

Figure A2: Unchanged label rates

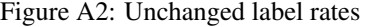

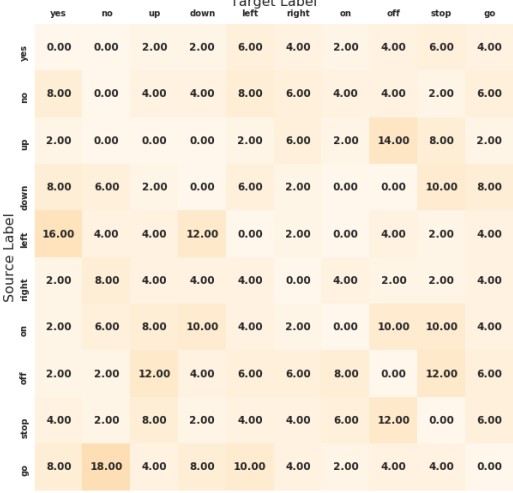

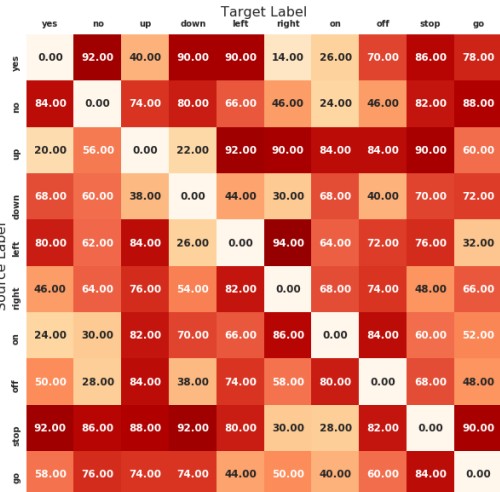

Figure A3: Successful attack rates after transformation Figure A4: Unchanged label rates after transformation

## 5.3 MORE RESULTS ON ADAPTIVE ATTACKS AGAINST TEMPORAL DEPENDENCY BASED METHOD

Table A5: Examples of Segment Attack and Concatenation attack

| Type | Transcribed results |
|---|---|
| Original | and he leaned against the wa lost in reveriey |
| the first half of Original | and he leaned against the wa |
| | |
| Adaptive attack target | this is an adversarial example |
| Adaptive attack result | this is an adversarial losin ver |
| the first half of Adv. | this is a agamsa |
| | |
| Adaptive attack target | okay google please cancel my medical appointment |
| Adaptive attack result | okay google please cancel my medcalosinver |
| the first half of Adv. | okay go please |
| Original | why one morning there came a quantity of people and set to work in the loft |
| Attack target | this is an adversarial example |
| | |
| $S_k$ | this is an |
| $S_{k-}$ | adversarial example |
| $S_k + S_{k-}$ | this is a quantity of people and set to work in a lift |
| | |
| $S_k$ | this is an adversarial example |
| $S_{k-}$ | $sil$ |
| $S_k + S_{k-}$ | this is an adernari eanquatete of pepl and sat to work in the loft |

Table A6: AUC scores of different $k$

| $k$ | WER | CER | LCP |
|---|---|---|---|
| 1/2 | 0.930 | 0.933 | 0.806 |
| 2/3 | 0.930 | 0.948 | 0.826 |
| 3/4 | 0.933 | 0.938 | 0.839 |
| 4/5 | 0.955 | **0.969** | 0.880 |
| 5/6 | 0.941 | 0.962 | 0.858 |

Table A7: AUC of detecting Combination Attack based on TD method

| Combination Attack | Detection Parameter $k_D$ | TD metrics WER | CER | LCP |
|---|---|---|---|---|
| $k_A = \{\frac{1}{2}\}$ | 1/2 | 0.607 | 0.518 | 0.643 |
| | 2/3 | 0.957 | 0.965 | 0.881 |
| | 3/4 | 0.943 | 0.951 | 0.875 |
| | Rand(0.2, 0.8) | 0.889 | 0.882 | 0.776 |
| $k_A = \{\frac{2}{3}\}$ | 1/2 | 0.932 | 0.912 | 0.860 |
| | 2/3 | 0.611 | 0.543 | 0.604 |
| | 3/4 | 0.956 | 0.944 | 0.872 |
| | Rand(0.2, 0.8) | 0.879 | 0.890 | 0.762 |
| $k_A = \{\frac{1}{2}, \frac{2}{3}\}$ | 1/2 | 0.633 | 0.690 | 0.552 |
| | 2/3 | 0.536 | 0.615 | 0.524 |
| | 3/4 | 0.942 | 0.974 | 0.934 |
| | Rand(0.2, 0.8) | 0.801 | 0.880 | 0.664 |
| $k_A = \{\frac{1}{2}, \frac{2}{3}, \frac{3}{4}\}$ | 1/2 | 0.665 | 0.682 | 0.604 |
| | 2/3 | 0.653 | 0.664 | 0.564 |
| | 3/4 | 0.633 | 0.653 | 0.601 |
| | Rand(0.2, 0.8) | 0.785 | 0.832 | 0.642 |
| $k_A = \{\frac{1}{2}, \frac{2}{3}, \frac{3}{4}, \frac{4}{5}\}$ | 1/2 | 0.701 | 0.712 | 0.615 |
| | 2/3 | 0.684 | 0.701 | 0.583 |
| | 3/4 | 0.681 | 0.693 | 0.613 |
| | Rand(0.2, 0.8) | 0.742 | 0.811 | 0.623 |
| $k_A = \{\frac{1}{2}, \frac{2}{3}, \frac{3}{4}, \frac{4}{5}, \frac{5}{6}\}$ | 1/2 | 0.736 | 0.784 | 0.601 |
| | 2/3 | 0.723 | 0.763 | 0.612 |
| | 3/4 | 0.715 | 0.755 | 0.584 |
| | Rand(0.2, 0.8) | 0.734 | 0.801 | 0.620 |
| $k_A = $ Rand(0.2, 0.8) | 1/2 | 0.880 | 0.881 | 0.824 |
| | 2/3 | 0.922 | 0.972 | 0.831 |
| | 3/4 | 0.952 | 0.968 | 0.894 |
| | Rand(0.2, 0.8) | 0.873 | 0.875 | 0.799 |

