# OpenReview forum: "Characterizing Audio Adversarial Examples Using Temporal Dependency"
_ICLR.cc/2019/Conference_

### Official Review · AnonReviewer1 · 2018-11-02

**Rating:** 7
**Confidence:** 3

**Review:**

This paper investigates adversarial examples for audio data. The standard defense techniques proposed for images are studied in the context of audio. It is shown that these techniques are somewhat robust to adversarial attacks, but fail against adaptive attacks. A method exploiting the temporal dependencies of the data is then presented and shown to be robust to adversarial examples and to adaptive attacks.

The paper addresses an important issue, and the two main findings of the paper, the transformation methods used in computer vision are not useful against audio adversarial example and using temporal dependencies improves the defense capability are significant. The proposed TD method is novel.

The first part of the paper is easy to read (Section 1-3), but Section 4 is hard to follow, for the following reasons:
* Section 4.1 presents the metrics used in the evaluation, which is nice. But in the following subsections, other metrics are used: effectiveness ratio, detection rate and relative perturbation. They should be clearly defined in 4.1, and the authors should discuss why they used these metrics.
* Section 4.2 should be reorganized as it is hard to follow: there are three types of attack, so one subsection per attack should make the section clearer.
* It's not always clear what each attack is doing and why it is used. I suggest the authors to have a separate subsection with the description of each attack and the motivation of why it is used.

Because of the above, it's hard to clearly assess the performance of each method for each attack, it would be better to have a Table that summarizes the results for the transformation methods. Also, I don't understand the statement in 4.2: "We report that the autoencoder works fine for transforming benign instances (57.6% WER in Common Voice compared to 27.5%)": given that it's not an attack, the PER should be the same with and without transform, as we don't want the transform to affect non-adversarial examples ? Please clarify that.
The experiments on the proposed TD method are clear enough to show the viability of the approach.

Overall, the findings of this paper are significant and it is good step towards audio adversarial examples defense. But the experimental part is hard to follow and does not bring a clear picture. I am still willing to accept the paper if the authors improve and clarify Section 4.

Revision after rebuttal:
The new version is definitely clearer and easier to read, hence I support the paper for acceptance and change my rating to 7.
There are still minor improvements that can be done in Section 4 to improve the overall clarity:
* About the metrics, the "Average attack success rate" and the "Target command recognition rate" should be clearly defined, probably under the description of the attack methods.
* The Adaptive attack approach could be introduced unter "Attack methods" in 4.1.
* Table 4 is not easy to read, the authors should improve it.
* The first paragraph in Section 4 ("The presentation flows ...") is very interesting, but almost reads like a conclusion, so maybe the authors could move that to the end of Section 4 or to Section 5.

---

> ### Author Response · Authors · 2018-11-26
> **Our response to Reviewer 1**
>
> Thanks for your insightful comments.
>
> Q: hard to understand Section 4:
> A: We apologize that we did not put enough efforts in presenting the experimental results in Section 4. Based on the review comments, we have reorganized and revised Section 4 to make the presentation clearer, including adding new tables  (Tables 1 & 4) that highlight the overall structure of our attack & defense / detection.
>
>
> Q: Also, I don't understand the statement in 4.2: "We report that the autoencoder works fine for transforming benign instances (57.6% WER in Common Voice compared to 27.5%)": given that it's not an attack, the WER should be the same with and without transform, as we don't want the transform to affect non-adversarial examples ?
> A: We agree with the reviewer that in this setting an “ideal” autoencoder would not affect the performance of benign examples and will mitigative the negative effects of adversarial examples. However, in our experiments, we were not able to find such an ideal autoencoder.
> Given the reconstruction nature of autoencoder based on the training data, here we aim to do an ablation study to make sure that the applied transformation will not affect the translation results of benign instance too much. And there appears to be a tradeoff between accuracy and robustness.

---

### Official Review · AnonReviewer2 · 2018-11-07
**Important problem, reasonable evaluation, hard to follow.**

**Rating:** 6
**Confidence:** 3

**Review:**

This paper presents a study of the problem of generating adversarial examples for speech processing systems. Two versions of this problem are considered: attacks against audio classification and against text to speech. The authors first study a the practice of input transformations as means of defense against adversarial examples. To do so, they evaluate three recent adversarial attacks on audio classification and TTS models trained on several datasets. It is found that input transformations have limited utility against adaptive attacks. Moreover, a novel type of defense is developed in which the prefix (of some fixed length) of the audio input is converted to text and compared with the prefix of the text output of the entire input, flagging the input as adversarial if sufficient mismatch is detected. It is found that this method is robust against a number of attacks.

This paper tackles a relevant problem and presents some surprising (the robustness of the prefix method) as well as some not surprising results. The evaluation has reasonable enough breadth to give the conclusions credibility.

My main complaint is that the exposition is somewhat hard to follow at places, especially in section 4. It is hard to keep track of which attack is applied to which scenario and what the conclusions were. Perhaps this could be summarized in some kind of high-level table. It would also be greatly beneficial if the attacks are briefly summarized somewhere. E.g., without following the references, it is completely unclear what is the "Commander Song" setting and what is it important.
Finally, I would advise the authors to not use the term "first k portion". This made understanding their proposed defense much harder than it needed to be. Perhaps "prefix of length k" or something along these lines would be easier to follow.

In summary, if the authors commit to improving the clarity of the paper,  I would be willing to support its acceptance by virtue of the breadth of the investigation and the importance of the problem.

---

> ### Author Response · Authors · 2018-11-26
> **Our response to Reviewer 2**
>
> Thanks for your constructive comments.
>
> Q: writing in Section 4
> A: We apologize that we did not put enough efforts in presenting the experimental results in Section 4. Based on the review comments, we have reorganized and revised Section 4 to make the presentation clearer, including adding new tables (Tables 1 & 4) that highlight the overall structure of our attack & defense / detection.
>
>
> Q: prefix of length k
> A: Thanks for the precious suggestion and we modified the name as you suggested in the revised version.

---

### Official Review · AnonReviewer4 · 2018-11-12
**Interesting findings but hard to fully understand the experiments.**

**Rating:** 6
**Confidence:** 3

**Review:**

This paper proposed a study on audio adversarial examples and conclude the input transformation-based defenses do not work very well on the audio domain, especially for adaptive attacks. They also point out the importance of temporal dependency in designing defenses which is specific for the audio domain. This observation is very interesting and inspiring as temporal dependency is an important character that should be paid attention to in the field of audio adversarial examples. They also design some adaptive attacks to the defense based on temporal dependency but either fail to attack the system or can be detected by the defense.

Based on the results in Table S7, it seems like being aware of the parameter k when designing attacks are very helpful for reducing the AUC score. My question is if the attacker uses the random sample K_A to generate the adversarial examples, then how the performance would be. Another limitation of this work is that the proposed defense can differentiate the adversarial examples to some extent, but the ASR is not able to make a right prediction for adversarial examples. In addition, the writing of Section 4 is not very clear and easy to follow.

In all, this paper proposed some interesting findings and point out a very important direction for audio adversarial examples. If the author can improve the writing in experiments and answer the above questions, I would support for the acceptance.

---

> ### Author Response · Authors · 2018-11-26
> **Our response to Review 4**
>
> We appreciate your insightful comments and feel sorry about hard following in Section 4. Here are some response to questions you concerned and we’ve uploaded new version of our paper with clearer structure.
>
> Q: My question is if the attacker uses the random sample K_A to generate the adversarial examples, then how the performance would be.
> A: Following your comment, we added the corresponding experiments in TableA7 in Appendix when k_A = rand(0.2, 0.8). We observe that even k_A is chosen randomly, the results are similar to k_A equals to a fixed number when k_D is also a random number. And when k_D is a fixed number, the attack detection results are also good because if k_A is not close to defender’s k_D, the attack effectiveness will be limited.
>
> Q: Another limitation of this work is that the proposed defense can differentiate the adversarial examples to some extent, but the ASR is not able to make a right prediction for adversarial examples
> A: Yes, the proposed TD method is a detection instead of defense method, and our goal is to tell the adversarial instances apart from benign. In many scenarios, detection is very important. For instance, in malware or adversarial audio based attacks, if users can detect adversarial instances and remove or ignore them, it indeed helps to ensure system security.
>
> Q: writing in Section 4
> A: We apologize that we did not put enough efforts in presenting the experimental results in Section 4. Based on the review comments, we have reorganized and revised Section 4 to make the presentation clearer, including adding new tables (Tables 1 & 4) that highlight the overall structure of our attack & defense / detection.

---

### Public Comment · (anonymous) · 2018-10-24
**"Attack cannot succeed"**

You write that the adaptive attack "cannot succeed". I wonder if you have tried verifying that the attack does eventually succeed if you allow larger distortions as suggested by (Athalye et al. 2018) to verify the attack is working as expected.

---

> ### Author Response · Authors · 2018-10-25
> **Reply to "Attack cannot succeed"**
>
> Thank you for the valuable comments.
> Yes, we indeed allow the distortion to be the largest possible in practice, which is the amplitude of the original audio waveform. If the distortion is larger than the original audio waveform, the benign audio will be totally covered by the adversarial one and hence the adversarial attack will become trivial but meaningless.
>
> Such rational setting for maximum distortion is also mentioned in (Carlini et al. 2018), and we will add corresponding discussion in our updated version as well. Thank you for pointing this out.

---

### Public Comment · (anonymous) · 2018-10-26
**example for TEMPORAL DEPENDENCY BASED METHOD**

Hi, for TEMPORAL DEPENDENCY BASED METHOD, you only show the example that perturbs a sentence to a totally different one without any same words. However, if the adversarial attack just changing one or two keywords in the sentence, will your method still be effective? For example, perturb from "Alex, call Tom and open the front door" to "Alex, call Tome and open the back door".

---

> ### Author Response · Authors · 2018-10-29
> **reply to "example for TEMPORAL DEPENDENCY BASED METHOD"**
>
> Thank you for the interesting question.
> First, in this work we aim to detect the state-of-the-art attacks such as [Carlini et al], and therefore we obtain their adversarial examples online directly and perform detection for the purpose of fair comparison.
> In addition, we conduct a new set of experiments these days based on your suggestion.
> In particular, we set the adversarial target as just modifying one single word, such as  it -> he, morning -> evening, open -> close, and perform the TD detection against them (we list some examples in the end of this answer). Overall, for 50 samples, if we choose a single k, the AUC based on WER and CER score are around 0.65, and LCP 0.7.
> We also try to perform TD based on ensemble of k as k = (3/8, 4/8, 5/8, 6/8, 7/8} and select the maximum score among WER / CER / LCP for each k, which leads to AUC =0.832.
> This shows that TD still performs effectively for detecting such “single word” adversarial audio, though less powerful than detecting those with larger perturbation.
> The reason is obvious by looking at the examples below. In each example, row 1 shows the translation result of the first k portion (TD), row 2 shows the “single word” adversarial target (Adv), while row 3 denotes the original benign translation (Benign).
> We can see that the perturbation of such “single word attack” will only introduce inconsistency for the “target word”, and therefore reduce the gap between adversarial and benign instances. Though harder to detect, such attacks limit the adversary’s attack ability in practice. If we want to detect them more effectively, better similarity metrics may be helpful, and we will explore such metrics in our future work.
>
> Examples:
>
> TD: and he [leges] against the wa lost in r
> Adv: and he [fights] against the wa lost in reveriey
> Benign: and he [leaned] against the wa lost in reveriey
>
> TD: this [perning] they all
> Adv: this [morning] they all said
> Benign: this [evening] they all said
>
> TD: they [eafned] about the fur tree and rustled among the bra
> Adv: they [learned] about the fur tree and rustled among the branches
> Benign: they [snuffed] about the fur tree and rustled among the branches
>
> TD: in the [voning] the servants and the housemaid kape
> Adv: in the [evening] the servants and the housemaid came in
> Benign: in the [morning] the servants and the housemaid came in
>
> TD: said the fur [ouse] thinking over what he had himself
> Adv: said the fur [mouse] thinking over what he had himself related
> Benign: said the fur [tree] thinking over what he had himself related
>
> TD: the [erfficer] as well as the young ladies te
> Adv: the [officer] as well as the young ladies decparated it
> Benign: the [servants] as well as the young ladies decparated it

---

### Author Response · Authors · 2018-11-26
**General Response**

We thank the reviewers for their valuable comments and suggestions. We are happy to learn that the contributions of this are well acknowledged by all reviewers. Meanwhile, we are sorry that the reviewers feel the experiment part (Section 4) is hard to follow. Based on the review comments, we have reorganized and revised Section 4 to make the presentation clearer, including adding new tables (Tables 1 & 4) that highlight the experiment settings and the overall structure of the considered attack & defense / detection.

Specifically, we made the following updates to our revision:
1. We rewrite Section 4 to make the structure more clear. Previously, since there are too many defense and attack methods, some of which requires slightly different evaluation metrics, Section 4 can be confusing as pointed by reviewers. In our revision, we added Table 1 and Table 4 to illustrate the defense and attack methods we evaluated, as well as the corresponding evaluation metrics and brief result summary.
2. We added additional experiments to show that even when K_A and K_D are both random, our proposed TD method can still detect different attacks with high AUC.
3. We also added additional experiments in Appendix Table A7 according to Review 4’s comments to show that even when k_A is random and k_D equals to ½ , ⅔, ¾, or a random number chosen from [0.2, 0.8], our proposed TD method can still detect different attacks with high AUC.

Please don’t hesitate to let us know if you have any additional comments.

---

### Meta-Review · Area_Chair1 · 2018-12-14
**Interesting findings about audio adversarial examples**

**Confidence:** 5
**Recommendation:** Accept (Poster)

**Metareview:**

The authors present a study characterizing adversarial examples in the audio domain. They highlight the importance of temporal dependency when defining defense against adversarial attacks.

Strengths
- The work presents an interesting analysis of properties of audio adversarial examples, and contrasts it with those in vision literature.
- Proposes a novel defense mechanism that is based on the idea of temporal dependency.

Weaknesses
- The technique identifies adversarial examples but is not able to make the correct prediction.
- The reviewers raised issue around clarity, but the authors took the effort to improve the section during the revision process.

The reviewers agree that the contribution is significant and useful for the community. There are still some concerns about clarity, which the authors should consider improving in the final version. Overall, the paper received positive reviews and therefore, is recommended to be accepted to the conference.